# An Attempt to Assess the Impact of Pandemic Restrictions on the Lifestyle, Diet, and Body Mass Index of Children with Endocrine Diseases—Preliminary Results

**DOI:** 10.3390/nu14010156

**Published:** 2021-12-29

**Authors:** Agnieszka Zachurzok, Małgorzata Wójcik, Aneta Gawlik, Jerzy B. Starzyk, Artur Mazur

**Affiliations:** 1Department of Pediatrics, Faculty of Medical Sciences in Zabrze, Medical University of Silesia, 40-752 Katowice, Poland; 2Department of Pediatric and Adolescent Endocrinology, Chair of Pediatrics, Jagiellonian University Medical College, 31-000 Krakow, Poland; jerzy.starzyk@uj.edu.pl; 3Department of Pediatrics, Pediatric Endocrinology and Diabetes, School of Medicine in Katowice, Medical University of Silesia, 40-752 Katowice, Poland; agawlik@sum.edu.pl; 4Department of Pediatrics, Pediatric Endocrinology and Diabetes, Medical Faculty, University of Rzeszów, 35-601 Rzeszów, Poland

**Keywords:** COVID-19, obesity, children, physical activity

## Abstract

Background: Home isolation during the coronavirus 2019 (COVID-19) pandemic lockdown strongly impacted everyday life, affecting, in particular, eating habits and everyday activity. The aim of this study was to analyze the impact of the pandemic on behaviors and subsequent changes in body mass index (BMI) in children from Southern Poland. Methods: The study included 206 participants (104 females and 102 males) with a complete analysis of 177 participants (96 females and 81 males) with a mean age of 12.8 ± 2.6 years admitted to three pediatric endocrinology clinics (Rzeszów, Kraków, and Katowice) due to simple obesity, type 1 diabetes mellitus, somatotropin pituitary deficiency on growth hormone replacement therapy, and other endocrine and metabolic disorders between June and September 2020. The study used a self-prepared questionnaire regarding eating habits, physical activity, screen time, and sleep before and during the lockdown. Anthropometric measurements were performed under clinical settings twice (before the pandemic in January–March 2020, and in June–September 2020). Results: During the lockdown, BMI z-scores increased over the whole group, especially in obese children (0.073 ± 0.18, *p* = 0.002). The number of children who declared low and high physical activity of more than 60 min per day declined from 41.2% and 18.6% to 31.1% and 6.2% (*p* = 0.03 and *p* < 0.001), respectively; sleep times over 8 h increased (46.9% vs. 60.4% *p* = 0.007); screen times over 5 h daily increased (14.7% to 46.9%, *p* < 0.001). Eating habits did not change significantly. Conclusions: Daily physical activity and sleep levels were affected by the pandemic leading to the increase of BMI, especially in obese patients with endocrine disorders. During the COVID-19 pandemic, forward-thinking strategies must be developed to prevent childhood obesity.

## 1. Introduction

On 4 March 2020, the first confirmed case of severe acute respiratory syndrome coronavirus 2 (SARS-CoV-2) infection was announced in Poland. On 16 March, schools and kindergartens closed and subsequent restrictions were introduced in Poland. Cultural establishments and shops in galleries were closed, parties and gatherings were banned, access to parks and beaches was limited, and playgrounds and sports fields were closed. Additionally, in early April, a temporary ban on access to forests was introduced. Studies conducted in many regions of the world affected by similar problems have shown that pandemic-related restrictions have been related unfavorably to movement behaviors of children and youth [1,2,3,4]. Regardless of location, sudden and radical changes in daily habits and lifestyles of the children and adults occurred with a drastic reduction of any form of socialization and physical activity [1,2,3,4,5]. Home isolation strongly impacted everyday life, affecting, in particular, eating habits and everyday behaviors [5]. Due to restrictions on free movement, school closures, and changing diets around the world, cases of “covibesity” have been reported in children and adolescents [6]. A recently published literature review by Stavridou et al. showed that the main and basic cause of an increase in obesity rates in children and adolescents was school closure, along with other coronavirus 2019 (COVID-19) restrictions that have disrupted the everyday routine of children, adolescents, and even young adults, leading to changes in their eating behaviors and physical activities [7]. Some studies additionally showed that during the COVID-19 pandemic, children had heightened anxiety during the lockdown period [1,2,3,4,5,6]. This negative effect was even more apparent in children with a low physical activity level [8]. The results of the large, prospective study conducted in the United States of America (US) suggest that children performed less physical activity and engaged in more sedentary behavior during the beginning of pandemic period as compared to the time before [9]. Children residing in urban areas and/or within small apartments are faced with even greater challenges due to limited space and opportunities for physical activity and, hence, they have been more susceptible to weight gain [10]. Low physical activity levels have been suggested to impact both body fat composition and appetite dysregulation [11]. The problem seems to be especially significant in children with excess body weight. It was noticed that a higher body mass index (BMI) and lower age were associated with an increase in unhealthy food consumption [4,11,12,13]. As was shown recently based on an online survey, in Poland, an increase in BMI was associated with a reduction in vegetable, fruit, and legume intake, leading to weight gain (almost 30%) [14]. Furthermore, it seems that these initially short-term changes in physical activity, sedentary behavior, and diet in reaction to the COVID-19 pandemic may become permanently entrenched, leading to increased risk of obesity, diabetes, and cardiovascular disease in children [9].

The aim of this study was to analyze the changes in physical activity, eating habits, sedentary behaviors, leisure screen time, and sleep in school-aged children and youth in Southern Poland during the pandemic and the impact of these changes on body mass.

Study hypotheses are as follows:

The pandemic has a negative impact on physical activity, eating habits, play, sedentary behaviors, leisure screen time, and sleep in school-aged children and youth with chronic endocrine disorders.The pandemic has a negative impact on body weight and BMI in school-aged children and youth with chronic endocrine disorders.

## 2. Materials and Methods

### 2.1. Study Design

The study was conducted in three tertiary pediatric endocrinology clinics in the south of Poland (Rzeszów, Kraków, and Katowice) among outpatients visiting endocrinology, diabetes, and metabolic disease units for treatment or management due to simple obesity, type 1 diabetes mellitus, somatotropin pituitary deficiency on growth hormone replacement therapy, and other endocrine and metabolic disorders (Table 1).

The inclusion criteria were as follows:Consent to participate in the study.Availability of documented measurements of body weight and height.

The exclusion criteria were as follows:Lack of consent to participate in the study.Lack of documented measurements of body weight and height.

The study used a questionnaire and anthropometric measurements. The anthropometric measurements were performed in all participants from January to March 2020 (before the pandemic) and again from June to September 2020 (during pandemic lockdown).

### 2.2. Participants

There were 206 participants and complete data allowing for a full analysis were obtained from 177 (96 females and 81 males) with a mean age of 12.8 ± 2.6 years (range 5.7–18.5 years).

### 2.3. Procedures

The study used a self-prepared questionnaire containing questions about eating habits, physical activity, screen time, and sleep pertaining to the child and the family before the pandemic and during the lockdown (see Appendix A). The questionnaire was filled out by parents together with children with unlimited time (average time of completion was about 10 min) during outpatient visits that took place between June and September 2020. The questionnaire contained two parts with the same sets of questions: the first regarding the period before lockdown, and the second regarding period during pandemic. There were two questions about time during the day dedicated for low and high physical activity, with scoring as follows: less than 30 min, >30–60 min, >60–120 min, and more than 120 min. One question was about sleep time (scoring: less than 6 h, >6–8 h, >8–10 h, and more than 10 h). There was one question regarding the number of meals (scored 1–5 and more), and five questions about the frequency of consumption of sweet snacks, soft drinks, vegetables, fruits, and fast-food dishes. There was also a question about screen time (scoring: less than 1 h, >1–2 h, >2–4 h, >4–6 h, and more than 6 h). Additionally, in the part dedicated to pandemic period, there was a question about time dedicated for e-learning. The questionnaire also included 8 questions to assess the changes that occurred during the lockdown period compared to the period before the pandemic. They concerned the child’s physical activity, screen time, sleep time, nutrition, as well as the parent’s physical activity, screen time, sleep time, and nutrition. The questionnaire was pre-tested in a pilot study in a group of 20 people to verify that the questions were understandable. Any suggestions from the respondents were included in the final version of the questionnaire.

#### Anthropometric Measurements

Anthropometric measurements were performed in all participants from January to March 2020 (before the pandemic) and again from June to September 2020 (during pandemic lockdown). The mean period between the first and second measurements was 183 ± 24 days. Body weight was measured to the nearest 0.1 kg on a calibrated balance beam scale and body height was measured to the nearest 0.1 cm. BMI was calculated by dividing weight (in kilograms) by the square of height (in meters). The value of BMI was interpreted, and BMI z-scores were calculated according to World Health Organization standards. All measurements were performed by trained health care professionals (nurses/physicians) with professional, validated equipment.

### 2.4. Ethics

The study was conducted according to the Helsinki declaration and approved by the Ethics Committee of The Medical University of Silesia on 3 July 2020 (PCN/0022/KB1/117/20). 

### 2.5. Statistics

A statistical analysis was performed using Statistica 13.3 PL. The quantitative variables were described with the mean (SD, standard deviation), and qualitative variables were defined by frequency (%) for descriptive statistics of the participants’ baseline and lockdown characteristics. Normal distribution of data was checked with the Kolmogorov–Smirnov test (K-S test). We used a chi-square test for qualitative data comparison of variables in relation to pandemic situation (before and during lockdown), and dependent sample Student’s t test for quantitative comparison of variables in relation to pandemic situation (before and during lockdown), as appropriate. A *p*-value less than 0.05 was considered statistically significant.

## 3. Results

### 3.1. Descriptive Analysis

Descriptive data regarding children and their residential characteristics (urban/rural area, availability of a garden area) are presented in Table 2. Most of the studied children live in rural areas and in small towns (*n* = 120; 68.2%) with garden area availability (*n* = 128; 72.7%).

### 3.2. Anthropometric Data

We observed that during lockdown, BMI z-scores of children increased significantly (0.07 ± 0.34; Table 3), especially in males (0.094 ± 0.36). In females, the increase was insignificant. BMI z-scores increased during lockdown in 95 (53.7%) children and decreased in 82 (46.3%) children (*p* > 0.05). When we compared obese and non-obese subjects, an increase in the BMI z-score was present in both subgroups; however, only in obese children was there a significant increase (0.073 ± 0.18). The number of children with obesity increased from 76 (42.9%) prior to the pandemic to 80 (45.2%) during the lockdown (*p* > 0.05), whereas the number of children with BMI <85 percentile decreased from 81 (45.8%) to 76 (42.9%) subjects. Overweight was observed in similar number of children before and during lockdown (*n* = 20, 11.3% and *n* = 21, 11.9%), respectively.

During the lockdown, seven children changed their status from non-obese to overweight or obese and one changed from overweight to obese, whereas in two children we observed the conversion from obese to overweight and in another two from overweight to normal weight.

### 3.3. Physical Activity

The amount of time declared by participants on low physical activity was significantly reduced during the pandemic over the whole group (*p* = 0.002), but the reduction was greater for obese children (Table 4.). The amount of children who declared low physical activity of more than 60 min per day declined from 73 (41.2%) children before lockdown to 55 (31.1%) children during the pandemic (*p* = 0.03; Figure 1). Additionally, the declared high physical activity time decreased significantly during the pandemic over the whole group (*p* < 0.001); the reduction was also greater in the obese subgroup. Before lockdown, 33 children (18.6%) engaged in high physical activity for more than 60 min per day, and this decreased during lockdown to only 11 children (6.2%; *p* < 0.001; Figure 2). Most parents declared that their physical activity during lockdown did not change (88 parents 50%), whereas 47 parents (26.7%) reported a decrease in physical activity during lockdown, and 41 parents reported an increase (23.3%).

### 3.4. Sleep and Screen Time

The study revealed that during the pandemic, the length of sleep and screen time for children increased. During the lockdown, more children slept for more than 8 h than before (*n* = 107, 60.4% children vs. *n* = 83, 46.9% children; *p* = 0.007) and less slept less than 6 h (*n* = 4, 2.3% children vs. *n* = 13, 7.3% children; *p* = 0.02; Figure 3). A dramatic increase in children spending more than 5 h per day using computers and other electronic devices during the pandemic was noticed (*n* = 26, 14.7% children prior to and *n* = 83. 46.9% during the lockdown (*p* < 0.001); Figure 4).

### 3.5. Eating Habits

There were no differences in the number of meals per day, nor in the amount of snacks, sweets, soda, juices, vegetables, fruits, and fast-foods meals consumed before and during the lockdown over the whole group studied (*p* > 0.05).

## 4. Discussion

This study was designed to investigate the influence of the COVID-19 pandemic lockdown on body weight and lifestyle of *n* = 177 Polish children and adolescents from Southern Poland. We found significant increases in the BMI z-score over the whole group studied, which were especially pronounced in obese children. Moreover, a significant reduction in low and high physical activity was observed with the greatest degree of reduction among obese children. The lockdown was associated with an increase in screen and sleep time, but not a change in eating habits.

The COVID-19 pandemic has caused an unprecedented change in lifestyle and social relationships. The most vulnerable population of children and adolescents in early stages of physical, intellectual, and emotional development have been particularly impacted [1,13,15,16]. Understanding the impact of the COVID-19 lockdown on childhood obesity is of utmost importance due to the high burden of obesity in Poland and severe complications of COVID-19 linked to obesity. Following the first COVID-19 pandemic lockdown, the BMI z-scores of children increased significantly (0.07 ± 0.34), especially in males (0.094 ± 0.36). When we compared obese and non-obese subjects, an increase in BMI z-score occurred in both subgroups, however, only in obese children was the increase significant (0.073 ± 0.18). Similar findings were observed in other countries hard hit by the COVID-19 pandemic such as the U.S, China, and Italy [4,9,17,18]. A national online questionnaire for school-aged children was conducted in China before and after the COVID-19 lockdown [17,18]. In this study, overall BMI significantly increased from 21.8 kg/m^2^ to 22.6 kg/m^2^ after the lockdown [17,18]. Unhealthy weight gain in childhood is of long-term concern, because multiple studies show that obesity experienced in childhood is associated with overweight and obesity in adulthood [8,19,20,21,22]. For instance, obesity experienced at an age as young as 5 has been shown to be associated with significantly higher BMI to the age of 50 with higher fat mass at age 50 [8]. In our study group, we did not observe an overall significant increase in prevalence of obesity in children following the lockdown that was reported in China, the U.S and other countries [17,18,23]. However, the obese subgroup increased during the pandemic from 64 to 77 children. These differences may result from different group selection methods, research tools, and research location. We performed our study in hospital clinics, which could have an impact on the size of our study group and was a limitation of our study.

An increased in unhealthy behavior patterns during the COVID-19 lockdown period could explain our findings. In our cohort, a decrease in low and high physical activity was observed. In the obese subgroup, a significant reduction in low physical activity could be associated with an increase in BMI z-score. In Italy, Pietrobelli et al. observed that time spent on sports activities decreased, and sleep time increased significantly during the lockdown [13]. Additionally, in our study, sleep time increased and more children slept more than 8 h, which may be an advantage of the lockdown. Based on a national study in China, Jia et al. also reported a decrease in physical activity with an increase in sedentary activity during the COVID-19 lockdown [17]. We observed that during the pandemic, the number of children spending more than 5 h for screen time tripled. Simultaneously, similar to our results, a significant increase in screen time has been found by nearly 5 h per day since the lockdown, which was linked to childhood obesity in many other studies [13,17,23,24].

There were no differences in the number of meals per day, nor in the number of snacks, sweets, sodas, juices, vegetables, fruits and fast-food meals consumed before and during the lockdown over our studied group (*p* > 0.05). On the other hand, in a cross-sectional online survey of adult Polish citizens (*n* = 1097) conducted during the nationwide quarantine, found that increased BMI was associated with less frequent consumption of vegetables, fruit, and legumes and a higher consumption of meat, dairy, and fast-food meals [14]. In a longitudinal study among children in Italy, unhealthy eating including consumption of potato chips, red meat, and sugary drinks increased during the lockdown [13].

The different behaviors of children and adolescents among countries may be related to contextual factors such as the degree of restriction and the number of COVID-19 infections among countries that directly affected behavior. For example, during the lockdown in China, outdoor exercise was not allowed, whereas more mild restrictions ensued in some European countries [2,8,10,18]. Eating habits probably also had an impact as households tended to purchase ultra-processed, calorie-dense comfort foods. Data suggest that, in supermarkets, shelves that held crackers, chips, soda, sugary cereals, and processed ready-to-eat meals were depleted more quickly compared to the shelves that held flour, rice, and beans. While stocking up on shelf-stable food items is clearly a necessity for preparedness helping to minimize trips outside of the home, many children still experienced higher calorie diets during the pandemic period [25]. Our study was performed in hospital clinics, and we do hope that the consumption of healthy food in our patients was a result of our health care support.

The advantage of this study is that it used actual clinical diagnoses and measurements rather than self-reported data. Undoubtedly the strength and unique value of our research is that weight and height were measured by professionals using dedicated equipment. Therefore, the anthropometric measurements are reliable, without basic errors, which for obvious reasons burdens research that is based on online surveys.

The study has some limitations. First, the study’s generalizability is limited since it was conducted in three health centers in Poland, which had an impact on the size of the study group. The small size of the sample might not be representative of the general pediatric population in Poland. We realize that compared to other investigations, especially in large population studies, the size of our group is small. Many of participants were already obese and most of them have other comorbidities (Type 1 diabetes, Prader–Willi syndrome) that could be associated with obesity or abnormal eating habits requiring pharmacologic treatment. Due to the lack of available validated questionnaires dedicated to the assessment of the impact of a pandemic on children’s behavior at the time of designing the study, a self-prepared survey was used.

## 5. Conclusions

Daily physical activity and sleep levels were affected by the pandemic leading to the increase of BMI, especially in obese patients with endocrine disorders. During the COVID-19 pandemic, forward-thinking strategies must be developed to prevent childhood obesity.

## Figures and Tables

**Figure 1 nutrients-14-00156-f001:**
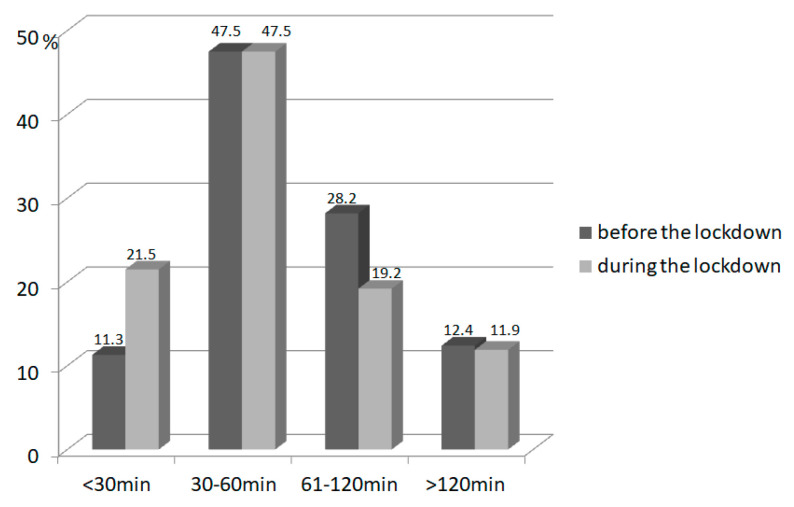
The amount of low physical activity (in minutes per day) in children (*n* = 177) before and during the lockdown.

**Figure 2 nutrients-14-00156-f002:**
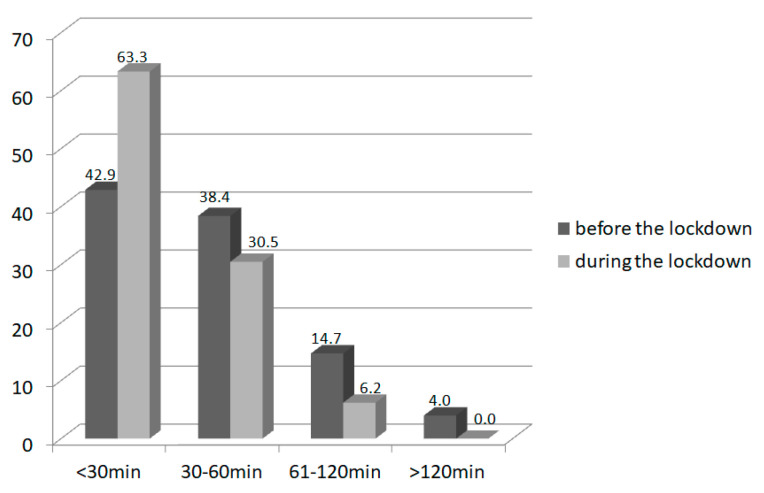
The amount of high physical activity (in minutes per day) in children (*n* = 177) before and during the lockdown.

**Figure 3 nutrients-14-00156-f003:**
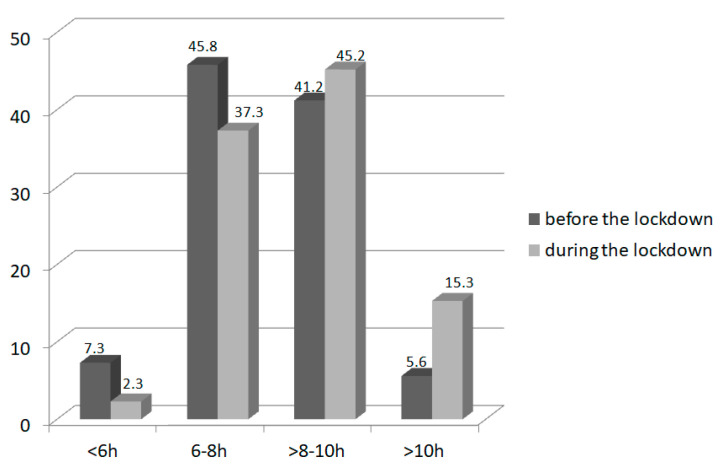
The length of the sleep (in hours per day) in children (*n* = 177) before and during the lockdown.

**Figure 4 nutrients-14-00156-f004:**
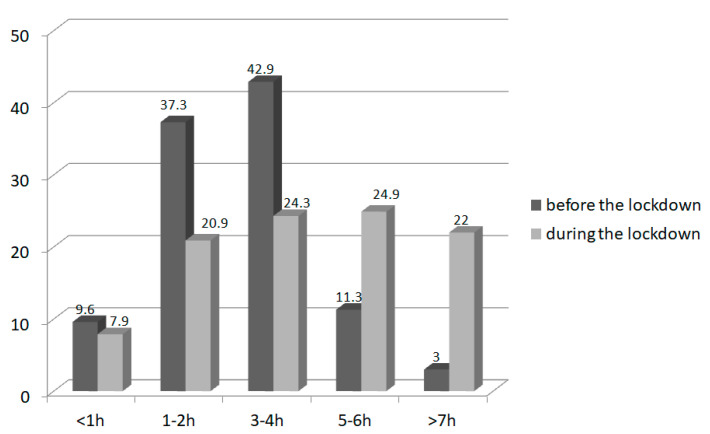
Screen time (in hours per day) in children (*n* = 177) before and during the lockdown.

**Table 1 nutrients-14-00156-t001:** The purpose of the visit in outpatient.

The Purpose of Visit in the Outpatient	Number	%
Obesity	64	36.2
Type 1 diabetes	36	20.3
Growth hormone deficiency	39	22.0
Thyroid disorders	10	5.6
Turner syndrome	7	4.0
Other endocrine disorders (growth and puberty disorders, Prader–Willi syndrome, SGA, gynecomastia)	15	8.5
Other lipids and carbohydrates metabolism disorders	6	3.4

SGA—small for gestational age.

**Table 2 nutrients-14-00156-t002:** Anthropometric data, the purpose of the visit in outpatient and residential characteristics during the lockdown.

	Total (*n* = 177)		Males (*n* = 96)		Females (*n* = 81)	
	Mean ± SD	Range	Mean ± SD	Range	Mean ± SD	Range
Age (y)	12.8 ± 2.6	5.7–18.5	12.9 ± 2.3	5.7–16.1	12.6 ± 2.8	6.8–18.5
Height (cm)	156.2 ± 17.5	111.7–195.0	160.7 ± 16.5	121.0–195.0	150.9 ± 17.1	11.7–183.7
Weight (kg)	59.8 ± 25.2	19.0–162.0	64.0 ± 26.2	20.6–162.0	54.9 ± 23.1	19.0–114.0
Garden space	Number	%				
None	48	27.3				
Small	16	9.1				
Large	112	63.6				
Place of residence	Number	%				
Village	93	52.8				
Small town	27	15.3				
Average size town	28	15.9				
Large city	28	15.9				

**Table 3 nutrients-14-00156-t003:** Changes in BMI z-score before and after the lockdown.

BMI z-Score	Before Lockdown (Mean ± SD)	During Lockdown (Mean ± SD)	Δ (Mean ± SD)	*p*
Study group (*n* = 177)	0.79 ± 1.35	0.86 ± 1.31	0.07 ± 0.34	0.006
Females (*n* = 96)	0.84 ± 1.34	0.89 ± 1.37	0.05 ± 0.32	0.1
Males (*n* = 81)	0.73 ± 1.36	0.82 ± 1.26	0.094 ± 0.36	0.02
Obese (*n* = 64)	2.02 ± 0.44	2.10 ± 0.38	0.073 ± 0.18	0.002
Non-obese (*n* = 113)	0.09 ± 1.18	0.16 ± 1.27	0.07 ± 0.40	0.064

Δ- change value.

**Table 4 nutrients-14-00156-t004:** Physical activity, computer usage, and dietary preferences before and during the lockdown.

Variable		Before Lockdown (Mean ± SD)	During Lockdown (Mean ± SD)	Δ (Mean ± SD)	*p*
Low physical activity (rated 1 to 4)	Study group (*n* = 177)	2.43 ± 0.85	2.22 ± 0.93	−0.20 ± 0.88	0.002
	Obese (*n* = 64)	2.47 ± 0.99	2.09 ± 1.06	−0.38 ± 0.97	0.003
	Non-obese (*n* = 113)	2.40 ± 0.76	2.29 ± 0.84	−0.11 ± 0.82	0.17
High physical activity (rated 1 to 4)	Study group	1.80 ± 0.83	1.43 ± 0.61	−0.37 ± 0.95	<0.001
	Obese	1.70 ± 0.90	1.28 ± 0.49	−0.42 ± 0.81	<0.001
	Non-obese	1.85 ± 0.79	1.51 ± 0.66	−0.33 ± 1.02	<0.001
Amount of sleep (rated 1 to 4)	Study group	2.45 ± 0.71	2.73 ± 0.74	0.28 ± 0.60	<0.001
	Obese	2.25 ± 0.64	2.53 ± 0.77	0.28 ± 0.74	0.004
	Non-obese	2.56 ± 0.73	2.85 ± 0.70	0.28 ± 0.49	<0.001
Screen time (rated 1 to 5)	Study group	2.66 ± 0.92	3.32 ± 1.25	0.66 ± 1.07	<0.001
	Obese	2.75 ± 0.93	3.36 ± 1.33	0.61 ± 1.23	<0.001
	Non-obese	2.61 ± 0.92	3.30 ± 1.21	0.69 ± 0.93	<0.001
Number of meals per day (rated 1 to 5)	Study group	4.25 ± 0.74	4.29 ± 0.71	0.04 ± 0.63	0.41
	Obese	4.26 ± 0.76	4.23 ± 0.85	−0,03 ± 0.56	0.66
	Non-obese	4.25 ± 0.74	4.33 ± 0.62	0.08 ± 0.67	0.21
Snacks/sweets consumption (rated 1 to 6)	Study group	2.64 ± 0.96	2.55 ± 1.08	−0.09 ± 0.75	0.11
	Obese	2.69 ± 0.99	2.66 ± 1.09	−0.03 ± 0.82	0.76
	Non-obese	2.61 ± 0.95	2.49 ± 1.07	−0.12 ± 0.71	0.065
Soda/juice consumption (rated 1 to 6)	Study group	2.29 ± 1.32	2.21 ± 1.27	−0.08 ± 1.0	0.29
	Obese	2.48 ± 1.29	2.31 ± 1.22	−0.17 ± 1.02	0.18
	Non-obese	2.18 ± 1.32	2.16 ± 1.30	−0.07 ± 0.99	0.77
Vegetable consumption (rated 1 to 6)	Study group	3.44 ± 1.10	3.34 ± 1.11	−0.10 ± 0.87	0.14
	Obese	3.33 ± 1.25	3.17 ± 1.24	−0.16 ± 1.10	0.26
	Non-obese	3.50 ± 1.00	3.44 ± 1.02	−0.06 ± 0.71	0.36
Fruit consumption (rated 1 to 6)	Study group	3.42 ± 1.08	3.42 ± 1.19	−0.006 ± 0.91	0.93
	Obese	3.39 ± 1.20	3.50 ± 1.31	0.11 ± 1.14	0.18
	Non-obese	3.44 ± 1.00	3.37 ± 1.12	−0.07 ± 0.75	0.21
Fast-food meals consumption (rated 1 to 6)	Study group	1.41 ± 1.11	1.36 ± 1.07	−0.05 ± 0.65	0.30
	Obese	1.52 ± 1.04	1.39 ± 0.94	−0.13 ± 1.06	0.35
	Non-obese	1.35 ± 1.14	1.34 ± 1.14	−0.01 ± 0.16	0.57

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
