# Peer review of "An Attempt to Assess the Impact of Pandemic Restrictions on the Lifestyle, Diet, and Body Mass Index of Children with Endocrine Diseases—Preliminary Results"

_nutrients, 2021, doi:10.3390/nu14010156_

Round 1

Reviewer 1 Report

Very interesting manuscript. Indeed you studied an important  population. Attached are my comments. The comments pertaining to the validity of you assessment tool, suggestions for additional analyses and figures are my most crucial comments.

Reviewer 2 Report

The present study tries to establish a relationship between the COVID-19 pandemic and the lifestyle habits of children and adolescents in Poland.

Abstract: OK.

Introduction: OK.

Materials and methods: From my point of view, the research is faulty since as an instrument for evaluating the variables, they use a questionnaire created by themselves, without being validated by any group of experts.

On the other hand, for anthropometric measurements they do not indicate the brand of the instruments used or their precision.

Results: the authors should explain a table and then insert it, not explain all the results and then put all the tables together.

Discussion: OK.

Conclusions: OK.

Informed Consent Statement: "Oral informed consent was obtained from all subjects involved in 287 the study".  An informed consent must always be read and signed on paper. It cannot be oral.

References:  underlined in some references (23)

Round 2

Reviewer 1 Report

Thank you for the detailed correction of the manuscript. You have fully respondent to all my comments. I think that the manuscript is much clearer now and I think that the readers of the journal will find it interesting.  

Reviewer 2 Report

I still think that first I should have validated the questionnaire that is used as an evaluation instrument. On this basis, a scientific article can never be accepted if the evaluation instrument has not been reviewed and accepted by the scientific community.